# Development of Lanzyme as the Potential Enzyme Replacement Therapy Drug for Fabry Disease

**DOI:** 10.3390/biom13010053

**Published:** 2022-12-27

**Authors:** Mulan Deng, Hongyu Zhou, Zhicheng Liang, Zhaoyang Li, Yanping Wang, Wanyi Guo, April Yuanyi Zhao, Fanghong Li, Yunping Mu, Allan Zijian Zhao

**Affiliations:** 1The School of Biomedical and Pharmaceutical Sciences, Guangdong University of Technology, Guangzhou 510000, China; 2The School of Medicine, South China University of Technology, Guangzhou 510000, China

**Keywords:** lysosomal storage disorders (LSDs), recombinant human α-galactosidase A (rhα-Gal A), Fabry disease (FD), enzyme replacement therapy (ERT), globotriaosylceramide (Gb3), Lanzyme

## Abstract

Fabry disease (FD) is a progressive multisystemic disease characterized by lysosomal enzyme deficiency. Enzyme replacement therapy (ERT) is one of the most significant advancements and breakthroughs in treating FD. However, limited resources and the high cost of ERT might prevent patients from receiving prompt and effective therapy, thereby resulting in severe complications. Future progress in ERT can uncover promising treatment options. In this study, we developed and validated a recombinant enzyme (Lanzyme) based on a CHO-S cell system to provide a new potential option for FD therapy. Our results indicated that Lanzyme was heavily glycosylated, and its highest activity was similar to a commercial enzyme (Fabrazyme^®^). Our pharmacokinetic assessment revealed that the half-life of Lanzyme was up to 11 min, which is nearly twice that of the commercial enzyme. In vivo experiments revealed that Lanzyme treatment sharply decreased the accumulation levels of Gb3 and lyso-Gb3 in various tissues of FD model mice, with superior or comparable therapeutic effects to Fabrazyme^®^. Based on these data, Lanzyme may represent a new and promising treatment approach for FD. Building this enzyme production system for ERT can offer additional choice, potentially with enhanced efficacy, for the benefit of patients with FD.

## 1. Introduction

Lysosomal storage disorders (LSDs) are a group of more than 50 severe metabolic diseases caused by the deficiency of specific lysosomal hydrolases, activators, carriers, or lysosomal integral membrane proteins, leading to the abnormal lysosomal accumulation of substrates [1]. In most cases, the mutation occurs on an enzyme for which functional deficiency leads to the accumulation of intermediate substrates, causing abnormal lysosomal activity due to a defective structural protein, ion pump, transporter, or cofactor [2]. LSDs often affect patients in early childhood, with an overall incidence of 1 in 5000 [3]. Various LSD disorders elicit highly variable symptoms and involve various factors, such as the age of onset and the mildness or severity of the disease. Their neurological signs and symptoms include developmental delay, movement disorders, seizures, dementia, deafness, and blindness. Other symptoms may include abnormal bone growth, gastrointestinal problems, enlarged livers (hepatomegaly), enlarged spleen (splenomegaly), dermatological, pulmonary, and cardiac issues, and ophthalmologic symptoms [4]. Late diagnosis or lack of treatment puts the patients at risk of developing significant, irreversible damage, loss of body functions, and life-threatening complications. Although most LSDs were described in the 20th century, the cellular pathogenesis underlying these diseases is complex and remains to be fully understood [4]. Thus, most of the treatment approaches developed to date for LSD treatment aim to increase the levels of missing enzymes in cells and tissues. These methods include hematopoietic stem cell transplantation, enzyme replacement therapy (ERT), pharmacological chaperone therapy, and gene therapy [5].

Fabry disease (FD) is an X-linked LSD caused by a deficiency in α-galactosidase A (α-Gal A; also referred to as GLA) activity. It is a progressive, life-threatening multisystemic disease caused by the intracellular accumulation of glycosphingolipids (mainly globotriaosylceramide [Gb3]) [6]. α-Gal A activity is essential for the efficient breakdown of Gb3, and reduced levels of this enzyme result in Gb3 accumulation in cells and organs throughout the body, contributing to life-threatening manifestations, such as kidney failure, heart disease, and stroke [7]. Since 2001, ERT has been available as a mode of treatment for FD and has provided significant clinical benefits to thousands of patients. Two preparations available, including agalsidase α (Replagal^®^), which is administered at a licensed dose of 0.2 mg/kg body weight, and agalsidase β (Fabrazyme^®^), used at a recommended dose of 1 mg/kg body weight [8]. Intravenous administration of both preparations is performed every other week. Patients in most European countries, as well as in Asia, Australia, and Canada, have access to agalsidase α and β. In 2003, the US FDA approved agalsidase β [9]. In some FD patients, ERT reduces Gb3 levels in plasma, urine, and microvascular endothelial cells, alleviates neuropathic pain, reduces hypertrophic cardiomyopathy, increases the ability to sweat, and stabilizes kidney function [10]. In addition, studies to uncover the role of the deacylated Gb3 analog globotriaosylsphingosine (lyso-Gb3) are becoming more common. Plasma lyso-Gb3 levels are markedly elevated in “classical” male FD patients compared with healthy individuals, and sometimes the levels are more than an order of magnitude higher than those of plasma Gb3 [11,12]. In symptomatic female FD patients, plasma lyso-Gb3 levels are significantly higher than normal, whereas plasma Gb3 levels remain in the normal range [13]. Most importantly, elevated plasma lyso-Gb3 levels correlate with an increased risk of cerebrovascular disease and left ventricular hypertrophy in male and female FD patients. Plasma lyso-Gb3 levels are also reduced by ERT, indicating the level of treatment efficacy [14].

ERT is one of the most significant advances and breakthroughs for treating LSDs [15]. This technique involves periodic intravenous infusions of human recombinant lysosomal enzymes produced and purified on a large scale from different sources using recombinant DNA techniques. Upon injection, the recombinant wild-type enzymes are taken up by tissues through the mannose 6-phosphate receptor (M6PR) or internalized by cells and targeted to the lysosomal compartment, thereby exerting therapeutic effects. In 1991, the FDA licensed the first ERT as an orphan drug, by approving alglucerase for the treatment of Gaucher disease (GD). More than two decades of clinical experience in treating thousands of patients helped to establish ERT as an effective therapeutic tool for alleviating the visceral, hematological, and biochemical manifestations of GD, accompanied by a substantial improvement in patients’ quality of life. The success of ERT in treating GD [16,17] has encouraged the use of this approach to treat the most prevalent lysosomal disorders [5]. To date, 16 more ERT drugs have been approved for GD type 1, and 10 other LSDs, such as Fabry disease and Pompe disease. Although ERT is currently being presently considered the standard of care for LSDs, its annual treatment costs are high, making it one of the most expensive drugs available on the market (the annual fee per patient is USD 200,000 in the US) [18].

LSDs have complex pathogenesis with a poor genotype–phenotype correlation, which has challenged research for prolonged periods. Due to the wide range of phenotypes in patients with LSDs, it can sometimes be difficult to diagnose these disorders, so misdiagnoses are common. Therefore, this phenomenon has caused pharmaceutical companies to often underestimate the number of LSD patients and misjudge market demand. This misestimation has led to the abandonment of various related research efforts, introducing a vicious circle. However, with the advancement of diagnostic technologies and better understanding of the disease, the incidence of LSDs is estimated to be far more than 1:10,000. For example, studies on FD have shown that the incidence is as high as 1:2000. The high annual treatment costs cause many patients to face unavailability of drugs, especially in developing countries, which hinders the diagnosis and treatment of LSDs. Therefore, reducing the annual treatment costs of LSDs is an urgent problem warranting immediate solutions. Consequently, this study aimed to establish an ERT treatment platform for LSD cases by using FD as an entry point to meet the needs of ERT treatment.

In the present study, we developed and validated a novel enzyme (Lanzyme) produced in a mammalian cell-based protein expression system and expressed in Chinese hamster ovary (CHO) cells, with the hope of solving this therapeutic issue. For further investigation, we chose to purify and characterize Lanzyme. Moreover, we performed a combination of in vitro and in vivo studies to assess the activity of Lanzyme, which showed an excellent half-life compared with some of the commercial enzymes (Fabrazyme^®^) [8], and successfully scavenged the accumulation of Gb3 and lyso-Gb3 in various tissues of an FD mouse model. Our findings suggest that Lanzyme may be a promising new therapeutic approach for FD treatment. Moreover, establishing an enzyme production system for ERT may lead to additional advantages for supporting rare disease therapies.

## 2. Material and Methods

### 2.1. Materials

A plasmid was constructed by inserting cDNA into the plasmid pcDNA4/HisMax-TOPO (Invitrogen, Carlsbad, CA, USA) and preserving it in our laboratory. ExpiCHO-S™ cells (Catalog No. A29127), ExpiCHO™ expression medium (Catalog No. A29100-01), and ExpiFectamine™ CHO transfection kits (Catalog No. A29129) were purchased from Thermo Fisher Scientific (Waltham, MA, USA). Rabbit monoclonal antibody (EP5828(2)) to galactosidase α was obtained from Abcam (Cambridge, UK). An α-galactosidase (α-GAL) activity assay kit (Catalog No. BC2570) was purchased from Solarbio (Solarbio, Beijing, China). A human α-galactosidase A/GLA enzyme-linked immunoassay (ELISA) kit (Catalog No. EH21RB) was purchased from Thermo Fisher Scientific. Fabrazyme^®^ (agalsidase-β) was obtained from Genzyme-Sanofi (Cambridge, MA, USA). Recombinant Human alpha-Galactosidase A/GLA Protein, CF (Rec. GLA, Catalog No. 6146-GH-020) was obtained from R&D Systems (Minneapolis, MN, USA). All drug solutions were prepared in accordance with the manufacturer’s instructions and recommendations. N-dodecanoyl-NBD-ceramide trihexoside and a fluorescent C12:0-ceramide trihexoside analog (ab143998) were obtained from Abcam (Cambridge, UK). Ceramide trihexoside (Gb3, Catalog #1067), lyso-ceramide trihexoside (lyso-Gb3, Catalog #1520), N-heptadecanoyl-lactosylceramide (Catalog#1538), and glucosylsphingosine (Catalog #1310) for liquid chromatography with tandem mass spectrometry (LC-MS/MS) analysis were purchased from Matreya (Pleasant Gap, PA, USA). Enzymes (endo H and PNGase F) for glycosylation analysis were purchased from New England BioLabs (Wicken, UK). Foreskin fibroblasts from FD patients were purchased from Coriell Institute (Camden, NJ, USA). β-Actin (#A5316, 1:5000 dilution) was purchased from Sigma (Saint Louis, MO, USA). MicroBCA Protein Assay Kit was purchased from Thermo. Water used throughout the study was either reverse-osmosis purified or high-performance liquid chromatography (HPLC) grade (Deventer, The Netherlands).

### 2.2. Enzyme Production

CHO-S cells were cultured in ExpiCHO™ expression medium (125-mL flask) and maintained at 37 °C, 8% CO_2_, and humidity saturation. The cell number was evaluated by a cell counter. The cells were passaged when the density reached 4 × 10^6^ cells/mL, and the passaged density was 0.3 × 10^6^ cells/mL. For transfection, the cell density was 7–10 × 10^6^ cells/mL diluted to 6 × 10^6^ cells/mL, and the pcDNA4-GLA plasmid was transfected in accordance with the instructions provided with the ExpiFectamine™ CHO transfection kit. An enhancer was added 18–22 h after transfection, and culturing was continued until the CHO-S cell viability was less than 75%. Then, the culture medium was collected, and the bioactivity of expressed proteins was further validated by Western blotting, and ELISA, and enzyme activity assay as described below.

### 2.3. Western Blotting and ELISA

Protein lysates were resolved by sodium dodecyl–sulfate polyacrylamide gel electrophoresis. After transblotting, the membranes were blocked with 5% skim milk for 1 h at room temperature and incubated with primary antibodies, followed by peroxidase-conjugated secondary antibody and chemiluminescent detection. The primary antibodies used were recombinant anti-galactosidase α (1:1000 dilution) and anti-β-actin (1:5000 dilution) antibodies. Quantitative data analysis was performed using ImageJ software. The expression of Lanzyme was quantified using a human α-Gal A ELISA kit. The protocol followed the kit’s instructions.

### 2.4. α-Gal Enzyme Activity Assay

An α-Gal activity assay kit for enzyme analysis was obtained from Solarbio (Guangzhou, China). Enzyme assays were performed in accordance according to the manufacturer’s instructions.

### 2.5. Glycosylation Analysis

Aliquots of the Lanzyme enzyme were digested with endo H and PNGase F following the manufacturer’s protocol. Endo H cleaves within the chitobiose core of high mannose glycan and cleaves some hybrid oligosaccharides from N-linked glycoproteins such as α-Gal A. PNGase F cleaves the innermost N-acetylglucosamine and asparagine residues of high mannose, hybrid, and complex oligosaccharides, removing nearly all N-linked oligosaccharides from the protein. Then, a 1:6 enzymatic hydrolysis solution and loading buffer were mixed, denatured at 95 °C for 10 min, and stored at −20 °C, and the glycosylation patterns of Lanzyme were verified by Western blotting.

### 2.6. Cellular Uptake and Gb3 Removal by Lanzyme

Fibroblasts from FD patients were seeded in 6-well plates at a density of 1 × 10^6^ cells in growth medium (Dulbecco’s modified Eagle medium with 10% fetal bovine serum) and incubated at 37 °C in 5% CO_2_ overnight. The cells were then incubated with Lanzyme (10 μg/mL) for 5 h, washed three times with a growth medium, and maintained in a growth medium at 37 °C and 5% CO_2_ for 2 days. After that, the cells were washed twice with Dulbecco’s phosphate-buffered saline and lysed (5 min in 200 µL of 0.5% Triton X-100, 27 mmol/L citric acid monohydrate, 46 mmol/L phosphate buffer, pH 4.6). The protein concentration in the lysates was determined using MicroBCA protein assay kit (Thermo Pierce, Waltham, MA, USA) in accordance with the manufacturer’s instructions. For Western blotting analysis in FD patients’ fibroblasts, 10–15 μg of total protein was loaded per lane and blotted as described previously. The net intensity of the α-Gal A band was normalized to the net intensity of the β-actin band in each lane. The ratio of α-Gal A to β-actin was compared across the lanes to calculate the relative increase in α-Gal A protein after coincubation. Enzyme activity was measured as described previously and was expressed as the nanomoles of pNP-Gal liberated per mg protein/h (nmol/mg protein/h).

### 2.7. Animal Procedures

Eight-week-old C57BL/6 male mice (WT mice) were purchased from the Model Animal Research Center of Nanjing University to evaluate Lanzyme pharmacokinetics and tissue and cellular uptake efficiency. FD mice (Gal-knockout mice) were purchased from Cyagen, and the male wild-type mice were mated with the female FD heterozygous mice to obtain heterozygous FD male mice for evaluating the therapeutic effect of Lanzyme. All animal experiments were conducted after approval from the Institutional Animal Care and Use Committee and the animal housing facility of the South China University of Technology (Guangzhou, PR China, Ethics permit number: 2021065). The mice were housed in plastic cages, at 30–70% humidity, at constant temperature (20 °C–24 °C), and 10:14 h light/dark cycle, with ad libitum access to a commercial rodent diet and drinking water.

### 2.8. Pharmacokinetics and Biodistribution

Enzyme preparations were injected at 1 mg/kg body weight (*n* = 8). Blood samples were collected by intraorbital puncturing at 0, 15, 30, and 45 min, as well as at 1, 1.5, and 24 h after the start of infusion. Plasma samples were centrifuged at 2700 g for 10 min at 4 °C and frozen at −80 °C until further activity assay and Western blotting analyses. The pharmacokinetic profile was determined by evaluating the enzymatic activity in plasma with a reagent test kit described above. Then, 24 h after the enzyme injection, the mice were perfused with saline (to remove the blood), and the heart, kidneys, spleen, and liver were harvested. One part of each organ was fixed with cold 4% paraformaldehyde, and the remaining part was immediately frozen in liquid nitrogen for further processing. Immunohistochemistry and Western blotting were used to detect tissue uptake of the enzyme.

### 2.9. Immunohistochemistry

The heart, kidney, liver, and spleen were harvested after enzyme infusion (1 mg/kg). Untreated male FD mice tissues were used as negative controls. The tissues were fixed in formalin, paraffin-embedded, and sectioned in 5-μm-thick slices. In brief, heat-induced epitope retrieval in EDTA buffer was performed, and the sections were treated with 3% hydrogen peroxide and 10% normal goat serum. The sections were incubated with rabbit monoclonal antibody against human α-Gal A (Abcam, Cambridge, UK). After incubation with a horseradish peroxidase (HRP)-labeled secondary antibody, signals were detected by 3,3′-diaminobenzidine (DAB) chromogen, and the sections were counterstained with hematoxylin. Signal specificity was verified with control staining, in which the primary antibody incubation was omitted. Compared with light and diffuse nonspecific staining in untreated controls, the specific signals displayed a granular cytoplasmic pattern, corroborating the findings from a previous study [19].

### 2.10. Tissue Gb3 and Lyso-Gb3 Clearance

Substrate reduction was evaluated in age-matched (18–22 weeks) FD mice, with eight males per group. The mice were intravenously injected with 1 mg/kg of Lanzyme four times, every other week. Neutral glycosphingolipids were then extracted from organs (heart, kidney, liver, and spleen) as follows: Tissue samples (5–15 mg) were homogenized in 600 μL methanol (containing an internal standard of 40 ng N-C17 and 40 ng glucosylsphingosine) to measure Gb3 and lyso-Gb3 hydrolysis. After adding 300 μL chloroform and 100 μL ddH_2_O, the mixture was shaken and then centrifuged for 10 min at 14,000 rpm. Finally, the samples were dried in a nitrogen blower and resuspended in 200 μL methanol for LC-MS/MS analysis [20].

### 2.11. Statistical Analysis

Statistical significance was determined using Excel 2020 (Microsoft, Redmond, WA, USA) or GraphPad Prism version 8 (San Diego, CA, USA), as specified in the figure and table legends. Linear trends for dose dependency were calculated using one-way ANOVA in GraphPad Prism. The half-life (t_1/2_) of recombinant human (rh)α-Gal A in plasma was calculated using a nonlinear one-phase exponential decay curve fitting function in GraphPad Prism [8]. Fluorescence, Western blotting, and IHC semi-quantitative analyses were conducted using Image J software.

## 3. Results

### 3.1. Production, Glycosylation, and Enzymatic Activity of Lanzyme

Lanzyme was transiently expressed in CHO-S cells. The expression of Lanzyme was assessed by Western blotting and ELISA. The results (Figure 1A) showed that Lanzyme was successfully expressed in CHO-S cells, and its expression was 100 ± 20 mg/L. To assess the glycosylation status of Lanzyme expressed by the CHO-S cells, the supernatants of CHO-S cells were digested with peptide-N-glycosidase F (PNGase F) and endoglycosidase H (Endo H). In the cell supernatant, Lanzyme displayed deglycosylation patterns similar to those of recombinant human α-Gal A enzymes produced in CHO cells (Figure 1B). To compare the activity of Lanzyme and Fabrazyme (the approved product), the enzymatic activity was measured. The results showed that Lanzyme and Fabrazyme had similar activities as 59 ± 9.1 U/mg and 57 ± 4.4 U/mg. The results also showed that the expression vector for α-Gal A recombinase had been constructed successfully in the CHO-S system. More importantly, Lanzyme underwent glycosylation, indicating that the expressed protein was mature. These findings indicated that Lanzyme was appropriately glycosylated, further supporting that the protein was expressed in the mature form.

### 3.2. Cellular Uptake and Gb3 Removal of Lanzyme

The evaluation of Lanzyme cellular uptake was performed by Western blotting and enzyme activity in FD patient-derived fibroblast cell lines. Lanzyme incubation produced greater α-Gal A levels and enzyme activity in FD fibroblasts compared with the control group. The results showed that Lanzyme was efficiently taken up into the foreskin fibroblasts of FD patients (Figure 1C). The Gb3 removal efficiency of Lanzyme was assessed using fibroblasts from FD patients. FD fibroblasts were pretreated with a fluorescent Gb3 analog (NBG-GB3), and then Lanzyme was added. Gb3 degradation activity was visualized using an inverted fluorescence microscope (Figure 1D). We found less yellow signal in the Lanzyme-treated group than in the control group, indicating effective targeting of lysosomes by Lanzyme and more effective clearance of NBG-Gb3.

### 3.3. Plasma Pharmacokinetics of Lanzyme

To investigate the in vivo effects of Lanzyme, 1 mg/kg Lanzyme was administered by intravenous infusion to 8-week-old male WT mice (*n* = 8). Plasma samples were collected over the following 24-h period, and α-Gal A activity was measured to determine the systemic exposure of active Lanzyme. As shown in Figure 2, Western blotting analysis revealed plasma Lanzyme levels. Activity assays determined the half-life of Lanzyme enzyme to be 11.8 min, compared with the commercial enzyme Fabrazyme with a half-life of approximately 7 min [8]. Our analysis revealed that the half-life of Lanzyme was substantially longer than that of Fabrazyme. Although the half-life of Lanzyme appears to be longer than that of Fabrazyme, further confirmation of this finding should wait until we measure the activity of Lanzyme derived from large-scale production for clinical use.

### 3.4. Tissue and Cellular Distribution of Lanzyme

Biodistribution analysis of Lanzyme at 24 h after injection (Figure 3A) revealed that a variety of tissues had effectively absorbed it. Moreover, the cellular distribution of Lanzyme was evaluated by Western blotting and immunohistochemistry. Eight-week-old male WT mice (*n* = 4) were administered a single intravenous bolus injection of 1 mg/kg Lanzyme, tissues were collected 24 h after administration, and immunohistochemical (IHC) staining was performed on paraffin-embedded sections using an antihuman α-Gal A antibody (Figure 3B). The extent of overall Lanzyme uptake was initially examined using ×20 original magnification (left panels). A specific α-Gal A IHC signal was not detected in untreated wild type mice (WT mice) in various tissues. Intravenous administration of Lanzyme yielded specific visible signals, demonstrating Lanzyme uptake into these tissues. Western blotting and IHC confirmed that various tissues effectively absorbed Lanzyme.

### 3.5. Biodistribution of Lanzyme in FD Mice

The Lanzyme ERT needs to be efficiently distributed between the different organs and tissues to exert its therapeutic effect. Twelve-week-old male FD mice were administered an intravenous infusion of 1 mg/kg Lanzyme (*n* = 8) or Fabrazyme (*n* = 12) (four biweekly administrations). Relevant tissues from FD mice, such as heart, kidney, liver, and spleen, were collected 7 days after injection, and the tissue enzyme activity levels of Lanzyme and Fabrazyme (measured by α-Gal A activity) were determined. We found significantly increased α-Gal A activity in all four tissues (Figure 4A).

IHC studies were conducted to evaluate the effect of Lanzyme on disease-relevant cell types. IHC staining was performed on paraffin-embedded sections using an antihuman α-Gal A antibody (Figure 4B). No specific α-Gal A IHC signal was detectable in the relevant tissues of the untreated FD mice (*n* = 9). Intravenous administration of Lanzyme and Fabrazyme generated readily visible signals for both, demonstrating that the uptake of Lanzyme was similar to that of Fabrazyme in these tissues. These results indicated that Lanzyme increased the total level of α-Gal A in specific disease-relevant cell types.

### 3.6. Substrate Reduction in Organs of Gal-Knockout Mice

Twelve-week-old male FD mice were administered Lanzyme (*n* = 8) or Fabrazyme (*n* = 12) as a positive control via intravenous bolus injection (1 mg/kg, four biweekly administrations) to determine the efficacy of Lanzyme in degrading accumulated Gb3 and lyso-Gb3 in vivo. The mice were then euthanized at 7 days after the final injection, with Gb3 and lyso-Gb3 levels measured in plasma, urine, and tissues by LC-MS/MS. Repeated administration of Lanzyme reduced Gb3 levels in plasma, urine, and tissues. The treated groups had significant reductions in Gb3 and lyso-Gb3 levels in their urine, plasma, and various tissues compared with the controls (Figure 5). Notably, the efficacy of Lanzyme in removing Gb3 and lyso-Gb3 was comparable to that of the commercial enzymes. Moreover, a prolonged blood circulation half-time of Lanzyme is likely improved the bioavailability of the drug. Nevertheless, it should not be disregarded that the drug–excipient interaction, such as compatibility, significantly affects bioavailability. Most importantly, our results suggest that Lanzyme might be a promising drug for the treatment of FD disease in the future.

## 4. Discussion

ERT is the only accepted therapy for FD to date [21]. In the early stage of research, researchers have extracted α-Gal A from the human placenta [22], hepatocytes [23], spleen, plasma [24], and fibroblasts [25], but is failed to meet the needs of FD treatment due to the low amounts of the extracted α-Gal A. α-Gal A is a glycoprotein, and its function is related to glycosylation. This phenomenon means that expression systems using *Escherichia coli* [26], insects [27], and yeast [28] have considerable shortcomings, such as low expression of recombinant human α-Gal A, imperfect posttranslational modification processing, and low activity of expression products. All these together render the expressed recombinant α-Gal A unsuitable for performing its role. In recent years, a nonphosphorylated form of A produced from moss [29] (referred to as moss-α-Gal) has been identified, which is endocytosed by the mannose receptor (MR). However, the role of MR in the therapeutic outcome of ERT can be α-Gal A–specific. The α-Gal A produced from tobacco cells (PRX-102), such as moss-α-Gal, is nonphosphorylated [30]. However, PRX-102 is chemically modified and forms a cross-linked dimer of PEGylated subunits. These modifications cause significant changes in protein characteristics, including different enzyme kinetics and dramatically prolonged circulation half-life (approximately 10 h) compared with agalsidase α or β. However, the uptake mechanism of PRX-102 remains to be elucidated. Moreover, the high annual treatment costs prevent many patients from obtaining the drugs, especially in developing countries, which further complicates and delays the diagnosis and treatment of FD. This study aimed to establish a universal protein expression platform for expressing various lysosomal enzymes, thereby promoting further investigations on LSDs to provide timely and effective drugs.

The glycosylation modification function of the CHO mammalian cell expression system is the same as that of human cells, which allows the expressed proteins to be the closest to natural protein molecules in terms of molecular structure, physical and chemical properties, and biological functions. The cells grow adherent to walls and are cultured in suspension, with high shear tolerance and osmotic ability. CHO cells efficiently amplify and express recombinant proteins with stable integration of foreign proteins, thereby allowing product exocytosis. The cells rarely secrete their own endogenous proteins, which is convenient for separating and purifying products downstream. High-density CHO cultures can be achieved by suspension culture or serum-free medium, and the culture volume can reach more than 100 L, which allows for large-scale production. The above advantages make CHO cells the first choice for a eukaryotic expression system [31,32,33]. Therefore, the CHO expression system was used in this experiment to express lysosomal enzymes.

In this study, the CHO-S suspension cell expression system was used to express α-Gal A, named Lanzyme, a new option for ERT for FD. We also aimed to verify the expression platform’s ability to provide ERT for LSDs with both efficacy and safety. Here, we characterized Lanzyme and performed its initial comparison to commercial ERTs. Our findings showed that CHO-S cells successfully expressed Lanzyme with intact enzymatic activity and glycosylation modifications, and the expression was as high as 100 ± 20 mg/L. Lanzyme targets lysosomes and efficiently removes NBG-Gb3 substrate analogs. Lanzyme plasma half-life is 11.8 min, allowing effective absorption by various tissues and removal of Gb3 and lyso-Gb3 in multiple tissues of FD mice so as to achieve the desired therapeutic efficacy, thereby preventing various complications of the disease, such as heart and kidney failure.

Protein glycosylation is one of the most important posttranslational modifications. It plays an essential role in protein biosynthesis and function by regulating protein folding, intracellular localization, stability, and solubility [34]. Although Lanzyme is efficiently expressed in CHO-S cells, its glycosylation still requires validation—a prerequisite for Lanzyme to perform its therapeutic role in FD. As predicted, our experimental results confirmed our hypothesis. Lanzyme demonstrated a very high recombinant human α-Gal A enzyme activity, similar to Fabrazyme. Moreover, adequate glycosylation also supported protein expression in its mature form. Tissues take up intravenously administered lysosomal enzymes through cell surface receptors that recognize the carbohydrate structure of the enzymes [29]. M6PR and MR represent two significant contributors to this uptake system. M6PR recognizes phosphorylated terminal mannose residues (M6P) and is expressed in most cell types [35]. It is generally believed that in ERT used for most LSDs, the M6PR-mediated endocytic pathway is crucial for sufficient enzyme delivery [36,37]. Based on this, we investigated whether fibroblasts from FD patients could take up Lanzyme and if it was scavenging the substrate analog NGB-GB3. The in vitro results demonstrated that Lanzyme was effectively taken up into cells and successfully cleared the target substrate from lysosomes, suggesting that Lanzyme had incontrovertible effects for halting the progression of FD, or even for curing it.

In pharmacokinetics, the half-life of plasma elimination indicates the time required for the plasma drug concentration to reduce by half. A short half-life is common in drugs with poor plasma stability, further impacting their distribution and efficacy. Lanzyme is rapidly metabolized, with a plasma half-life of approximately 11.8 min. Our findings interestingly revealed that the half-life of Lanzyme was almost twice as high as that of Fabrazyme (about 7 min) [8]. We hypothesized that the prolonged half-life might enable the drug to reach the target organs or cells better, indicating the potential for enhanced efficacy while reducing the amount of drug needed and the frequency of administration to achieve a therapeutic effect. A drug must have a controlled drug release profile to prolong its plasma half-life. Subsequently, we plan to continue refining our novel findings. Although the glycosylation constitutes 2–3% of the total mass of the protein, different glycoforms such as galactose, fucose, high mannose, N-acetylneuraminic acid (NANA), and N-glycoylneuraminic acid (NGNA) have considerable influence on the function of proteins, including their safety, immunogenicity, efficacy, biological activity, and clearance (pharmacodynamic and pharmacokinetic properties) [38]. However, the current research on protein glycosylation is challenging, warranting follow-up studies for conducting in-depth research to identify the glycosylation pattern.

The severe complications in the late stage of FD primarily include heart and kidney failure, and the disease progression directly correlates with the amount of storage substrates Gb3 and lyso-Gb3 [8]. Ceramide trihexose (CTH) is a glycosphingolipid in mammalian cell membranes [39]. The lack of α-Gal A-mediated conversion of CTH to lactosylceramide leads to the accumulation of CTH in lysosomes, which leads to FD development [40,41]. Therefore, the clinical diagnosis of FD mainly involves the detection of enzyme activity and the concentration of Gb3 in plasma and urine [41]. However, given that FD is an X-linked inherited disease, female patients with FD may have normal enzyme activity and plasma or urine levels, which makes diagnosis in females challenging [42]. Fortunately, the finding of deacylated ceramides in GD has helped to recognize lyso-Gb3 as a reliable diagnostic marker for female FD patients, who often have very high levels of lyso-Gb3 [42]. Several studies have confirmed that although the exact pathogenic mechanisms underlying renal damage in FD are yet to be identified, Gb3 deposition is the first step in a complex pathological pathway that leads nephrosclerosis and fibrosis [43]. Moreover, elevated lyso-Gb3 is associated with an increased risk of cerebrovascular disease and left ventricular hypertrophy, and lifetime exposure to plasma lyso-Gb3 is associated with disease severity [42]. Our results showed why Lanzyme effectively scavenged the accumulation of Gb3 and lyso-Gb3 in the plasma, urine, and tissues of FD mice. The finished products of the commercial enzyme and some adjuvants in Fabrazyme are related, which may promote the distribution and absorption of the drug. However, Lanzyme was produced under laboratory conditions and was purified only by HIS tag affinity chromatography, the purity might not be sufficient for practical function, resulting in a specific impact on the uptake of Lanzyme. Nevertheless, Lanzyme and Fabrazyme did not differ significantly in their substrate scavenging. Therefore, we plan continued include purifying the protein further and exploring the adjuvant composition and ratio of the commercial enzymes, we believe that the optimized enzyme may have more advantages in treating FD.

ERT changed FD’s natural progression for the first time [44]. There is a significant effect of ERT on plasma Gb3 levels, estimated glomerular filtration rate, and cardiometabolic outcomes. Additionally, ERT may relieve pain and improve quality of life, as well as the condition of the nervous and gastrointestinal systems. Notably, patients who received ERT at an earlier age had better outcomes. Therefore, prevention or mitigation of the disease requires effective early administration of a suitable treatment [45]. However, the infused enzymes tend to be unstable at neutral pH and body temperature, which mimic the physiological environment during infusion (i.e., the blood), resulting in a short circulating half-life of adequately folded, active enzyme in vivo [10]. Hence, there are multiple reasons for the very high costs of ERT [46], which lead to the termination of treatment by many patients who cannot afford the drug. Moreover, apart from the high cost of treatment, uncertainty in drug shortages such as those experienced during COVID-19 is a severe threat to the effective treatment of LSD patients [47]. Patients with more than 70 other LSDs can experience the same problem. Thus, drug accessibility for treating LSDs, especially that of the expensive ERT drugs, has become a critical problem that warrants in-depth investigation.

Solving the problem of the inaccessibility of LSD drugs requires joint efforts of many parties, such as the government and pharmaceutical companies. FD is one of the LSD-related ERTs that has been most extensively investigated. Biweekly infusion of recombinant α-Gal A ERTs, agalsidase β, or agalsidase α is the most suitable current treatment for FD. Its annual treatment cost is often in the millions dollars, and the incidence of FD was initially underestimated. The successful expression of Lanzyme in this study provides a new option for ERT for FD treatment, and we can provide low-cost, accessible drugs for patients. The successful expression of Lanzyme demonstrates that we have successfully constructed a mammalian cell expression platform focused on the expression of lysosomal enzymes to achieve initial success. Based on Lanzyme expression, we also expressed the enzyme defective in Pompe disease (acid α-glucosidase) and the enzyme defective in GD (glucocerebrosidase), which are currently in the early stage of research and development. We believe that there could be good expression efficiency. Therefore, this study provides a new option for ERT for FD treatment and establishes a novel LSD drug development platform. We aim to continue to research and develop other defective lysosomal enzymes in the future.

## Figures and Tables

**Figure 1 biomolecules-13-00053-f001:**
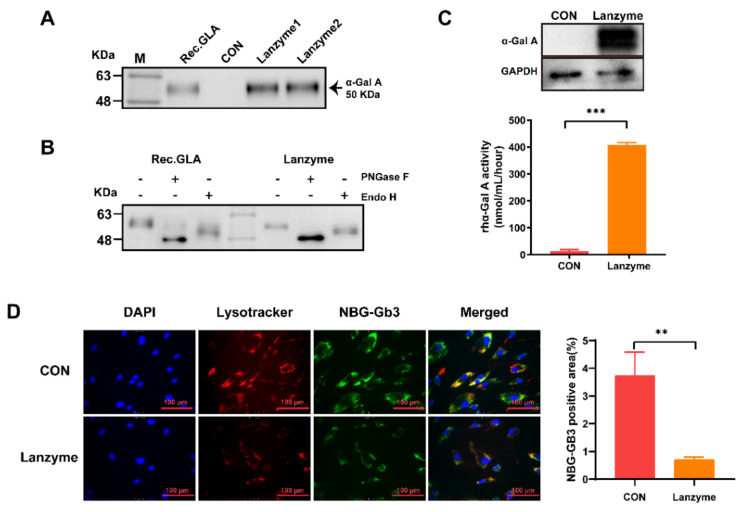
In vitro characterization studies of Lanzyme. (**A**) Lanzyme was expressed in CHO-S cells. M is the protein standard and molecular weights are shown on the left; Rec.hGLA produced in CHO cells is shown as a positive control; CON produced in CHO cells transduced with the pcDNA4 vector is shown as a negative control; Lanzymes 1 and 2 were produced from different batches in CHO-S cells. (**B**) The glycosylation status of Lanzyme. Representative Western blotting image showing Rec.hGLA and Lanzyme, with and without PNGase F or Endo H digestion. (**C**) Internalization of Lanzyme in FD fibroblasts. FD fibroblasts were incubated with Lanzyme (0.5 nmol/L) for 5 h. α-Gal A protein levels and enzyme activity in cell lysates were measured by Western blot and α-galactosidase A (α-Gal A) activity 2 days later. The data points α-Gal A enzyme activity shown are the three wells tested in parallel from one representative experiment. *** *p* < 0.001 compared to CON. (**D**) Internalization and lysosomal localization of Lanzyme in FD fibroblasts. FD fibroblasts were grown overnight on glass coverslips and incubated for 24 h in NBD-GB3 (10 μg/mL), then incubated for 5 h in the presence or absence of Lanzyme (4 μg/mL) and incubated for 24 h in a culture medium. DAPI (blue color) was used for staining the cellular nuclei. Note the colocalization of the lysosomal staining (red) and the NBG-Gb3containing granules (green). The overlap is represented in yellow when the images are superimposed. The data points shown are the three wells tested in parallel from one representative experiment. NBG-Gb3 semi-quantitative analyses were conducted using Image J software. ** *p* < 0.01 compared to CON.

**Figure 2 biomolecules-13-00053-f002:**
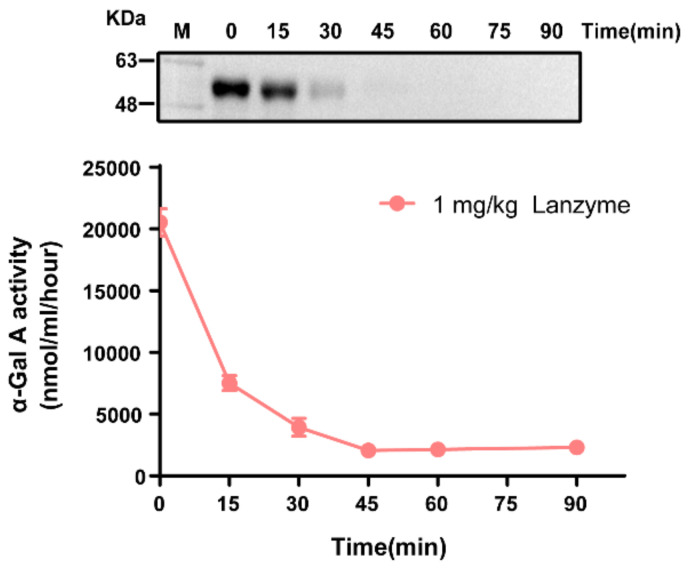
The circulating half-life of Lanzyme. Eight-week-old male wild-type WT mice (*n* = 8) were dosed with a single intravenous infusion of 1 mg/kg of Lanzyme. Plasma samples were collected over a 24 h period from the start of the infusion, and Lanzyme (α-Gal A) activity (lower panel) and protein levels (Western blotting, upper panel) were measured. For the graph, each time point represents the mean ± SEM of the activity measured from eight mice. Each lane of the Western blotting represents a different time point plasma from a single mouse.

**Figure 3 biomolecules-13-00053-f003:**
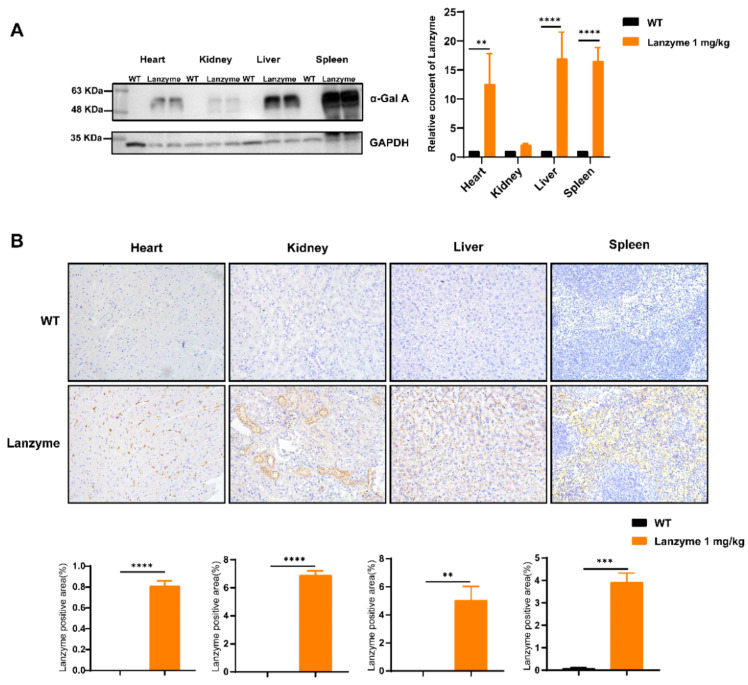
Tissue and cellular distribution of infused Lanzyme injected into WT mice. Eight-week-old male WT mice (*n* = 4) were administered an intravenous bolus injection of 1 mg/kg Lanzyme. Tissues were collected at 24 h after administration. Western blotting verified the uptake of Lanzyme in mouse tissues after administration (**A**), and IHC staining was conducted on paraffin sections using an antihuman α-Gal A antibody (**B**). Western blotting and IHC of Lanzyme in WT mice tissue semi-quantitative analyses were conducted using Image J software. * Comparison between WT mice, the uptake and distribution of Lanzyme in WT mices tissue. ** *p* < 0.01, *** *p* < 0.001, **** *p* < 0.0001.

**Figure 4 biomolecules-13-00053-f004:**
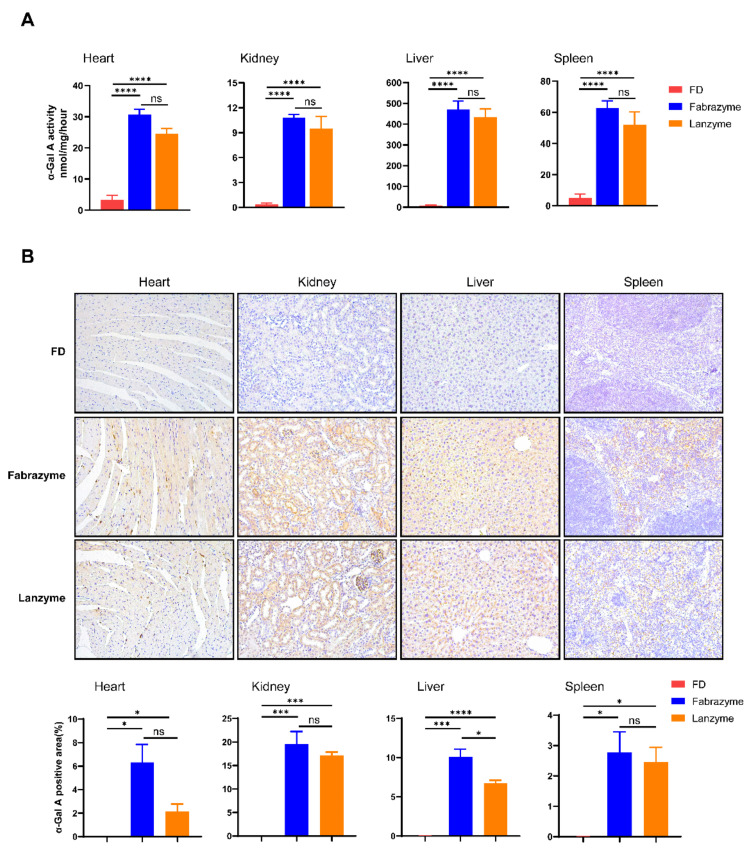
Lanzyme activity and tissue distribution in organs of FD mice. (**A**) Twelve-week-old male FD mice were administered an intravenous infusion of 1 mg/kg of Lanzyme. Heart, kidney, liver, and spleen were collected 7 days after infusion, and the activity of α-galactosidase A (α-Gal A) was determined. Each bar represents the mean ± SEM of five to six mice per group. **** *p* < 0.0001 compared with FD; ns * *p* > 0.05 compared to Fabrazyme; one-way ANOVA. (**B**) Twelve-week-old male FD mice were given an intravenous bolus injection of 1 mg/kg Lanzyme. Tissues were collected seven days after administration, and IHC staining was conducted on paraffin sections using an anti-human α-Gal A antibody. IHC of Lanzyme in WT mice tissue semi-quantitative analyses were conducted using Image J software. * Comparison between FD group. * *p* < 0.05, *** *p* < 0.001, **** *p* < 0.0001.

**Figure 5 biomolecules-13-00053-f005:**
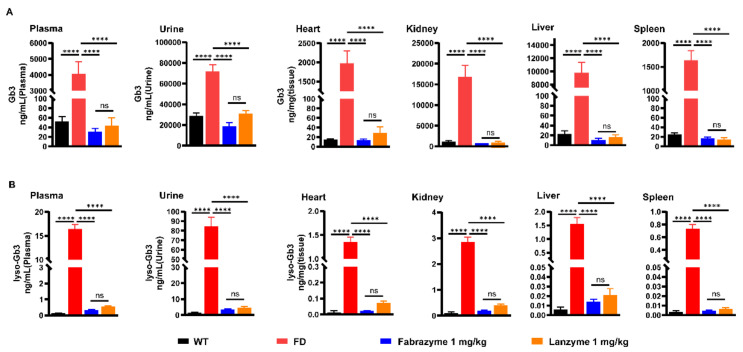
Efficacy of Lanzyme and Fabrazyme in clearing accumulated Gb3 (**A**) and lyso-Gb3 (**B**) in FD mice. Gb3 and lyso-Gb3 concentrations in plasma, urine, heart, kidney, liver, and spleen were analyzed 7 days after the last infusion of either Lanzyme or Fabrazyme at 1 mg/kg doses. Data are presented as mean ± SEM (*n* = 8). **** *p* < 0.0001, ns is non-significant. Statistical significance is shown on top of each Fabrazyme group and indicates no significant difference between Fabrazyme and the same dose of Lanzyme.

## Data Availability

The data that support the findings of this study are available on request from the corresponding author.

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
