# Peer review of "Development of Lanzyme as the Potential Enzyme Replacement Therapy Drug for Fabry Disease"

_biomolecules, 2022, doi:10.3390/biom13010053_

Round 1

Reviewer 1 Report

In this manuscript, the authors produced and tested the GLA enzyme, herein referred to as Lanzyme, for the ERT of Fabry disease. Although the evidence reported by the authors sounds attractive, several questions must be addressed before being considered for publication. Moreover, the manuscript presents numerous English mistakes or inappropriate language on several points. So, the English form should be fixed. Overall, an English-speaking person should revise the manuscript.

The following specific points need to be revised:

1.     The authors have declared that Lanzyme expression in CHO-S cells was 100 +/- 20 mg/L. How they got such a value?

2.     The authors have declared that Lanzyme and Fabrazyme had similar activities at 59 +/- 9.1 U/mg. How they got this value, and why was the result not shown in any figures?

3.     In Fig. 1D, the authors followed the fluorescence intensity of the Gb3 analog (NBG-Gb3) to verify the ability of Lanzyme to metabolize this analog and, as such, to confirm the Lanzyme uptake by the CHO cells and its delivery to the lysosomal compartment. However, in Lanzyme-treated cells, the lysotracker staining was significantly lower than the control, indicating a possible defect in lysosomal acidification. Why this effect? Moreover, the authors must show a control experiment for NBG-Gb3 clearance in wild-type fibroblasts. In addition, either the scale bar or a quantification graph, as well as any indications about the number of experiments performed, must be added.

4.     In Fig. 2, a loading control for the WB analysis is missing and must be added.

5.     In Fig. 3, a loading control for the WB analysis is missing and must be added. Moreover, a quantification graph for either WB or IHC analysis must be shown.

6.     Fig. 4: according to the enzymatic activity of GLA measured in each organ, the liver was the one with the highest value of enzyme uptake, followed by the spleen, then the heart, and finally by the kidney. However, as reported in Fig. 3A, the spleen was the highest, followed by the liver, then the kidney and finally by the heart. Why does this discrepancy?

Reviewer 2 Report

This is a well-written study that clearly and concisely evaluates a new recombinant enzyme for treating Fabry Disease via ERT. This study is logical and well-controlled.  I have some minor concerns about the following:

1) The sample size of each mouse cohort in the study should be indicated. 

2) For Figure 1, please clarify if the RecGLA is different than Lanzyme and the source of the RecGLA.

3) For Figure 5, the Plasma and Urine graphs are not labelled.

4) There is no direct comparison between the half-life of Fabrazyme and Lanzyme in the same mice, so it is difficult to know whether the difference in half-lives between the two enzymes are statistically significant.

5) You mention that Lanzyme does not reduce Gb3 or lyso-GB3 as well as Fabrazyme in some tissues, however, you stats indicate that the differences as ns. The distribution of Lanzyme is also slightly less than Fabrazyme in all of the organs indicated, but they are also ns, however, you don't mention this in your discussion. This seems a bit inconsistent.

6) In the discussion, you suggest that the glycosylation of Lanzyme is different than Fabrazyme and that this is the reason that Lanzyme has a prolonged half-life. How do you know this? You show no data to indicate that the glycosylation pattern between the two different enzymes is different.

Author Response

Q1) The sample size of each mouse cohort in the study should be indicated.

Response 1: We have added the sample size of each mouse cohort in the revised manuscript and highlighted with yellow color.

Q2) For Figure 1, please clarify if the RecGLA is different than Lanzyme and the source of the RecGLA.

Response 2: The RecGLA sequence is essentially the same as Lanzyme. RecGLA was purchased through R&D company, and we used it as a positive control for the analysis of glycosylation. We have added the source in the revised manuscript (page 3 line 136 and page 7 line 290).

Q3) For Figure 5, the Plasma and Urine graphs are not labelled.

Response 3: The Plasma and Urine graphs are now labelled. In Fig.5A, it is “Gb3 ng/mL” (Plamsa) and “Gb3 ng/mL (Urine)”. In Fig.5B, it is 'lyso-Gb3 ng/mL (Plamsa) and lyso-Gb3 ng/mL (Urine)'. Due to typesetting issues, the Gb3 and lyso-Gb3 statistical graphs of each organization need to share the Y axis, so the heart, kidney, liver, and spleen were labelled on the corresponding chart. (see the new Figure 5)

Q4) There is no direct comparison between the half-life of Fabrazyme and Lanzyme in the same mice, so it is difficult to know whether the difference in half-lives between the two enzymes are statistically significant.

Response 4: Indeed, at this stage, it is premature to do a straight comparison. Fabrazyme is an approved commercial enzyme, it’s prepared by the CMC process with a reported half-life of 7 minutes. Lanzyme is currently only in the laboratory developmental stage, and its half-life is 11.8 minutes. We have to admit that it is what is observed, nothing more than that. In the result section, we added a sentence, “Although the half-life of Lanzyme appears to be longer than that of Fabrazyme, further confirmation of this finding should wait until we measure activity of Lanzyme derived from large scale production for clinical use.”

Q5) You mention that Lanzyme does not reduce Gb3 or lyso-GB3 as well as Fabrazyme in some tissues, however, you stats indicate that the differences as ns. The distribution of Lanzyme is also slightly less than Fabrazyme in all of the organs indicated, but they are also ns, however, you don't mention this in your discussion. This seems a bit inconsistent.

Response 5: We sincerely thank the reviewer for the comment, and agree that Lanzyme and Fabrazyme did not differ significantly in their distribution and substrate scavenging. We have corrected this statement in the revised manuscript in page 13, line 488.

Q6) In the discussion, you suggest that the glycosylation of Lanzyme is different than Fabrazyme and that this is the reason that Lanzyme has a prolonged half-life. How do you know this? You show no data to indicate that the glycosylation pattern between the two different enzymes is different.

Response 6: We appreciate this comment. Indeed, there was no direct evidence in this study to prove that the difference in half-life of Fabrazyme and Lanzyme are associated with differential glycosylation. Therefore, the relevant statement had been deleted in the revised manuscript. In the follow-up experiment, we will carry out the experiments to explore this issue.

Round 2

Reviewer 1 Report

The authors have responded appropriately and satisfactorily to the comments and suggestions proposed by this reviewer.